# Frontostriatal Functional Connectivity Underlies the Association between Punishment Sensitivity and Procrastination

**DOI:** 10.3390/brainsci12091163

**Published:** 2022-08-30

**Authors:** Wenshan Dong, Jie Luo, Hangfeng Huo, Carol A. Seger, Qi Chen

**Affiliations:** 1Key Laboratory of Brain, Cognition and Education Sciences, Ministry of Education, Guangzhou 510631, China; 2School of Psychology, Center for Studies of Psychological Application, and Guangdong Key Laboratory of Mental Health and Cognitive Science, South China Normal University, Guangzhou 510631, China; 3Department of Psychology and Program in Molecular, Cellular, and Integrative Neurosciences, Colorado State University, Fort Collins, CO 80523, USA

**Keywords:** procrastination, punishment sensitivity, VBM, rsFC

## Abstract

Procrastination is defined as putting off an intended course of action voluntarily despite the harmful consequences. Previous studies have suggested that procrastination is associated with punishment sensitivity in that high punishment sensitivity results in increased negative utility for task performance. We hypothesized the effects of punishment sensitivity on procrastination would be mediated by a network connecting the caudate nucleus and prefrontal cortex, both of which have been previously associated with self-control and emotional control during procrastination. We employed voxel-based morphometry (VBM) and resting-state functional connectivity (rsFC) to examine the neural substrates of punishment sensitivity and its relationship with procrastination (N = 268). The behavioral results indicated a strong positive correlation between measures of punishment sensitivity and procrastination. The VBM analysis revealed that the gray matter (GM) volume of the right caudate was significantly positively correlated with punishment sensitivity. The primary rsFC analysis revealed connectivity between this caudate location and the bilateral middle frontal gyrus (MFG) was significantly negatively correlated with punishment sensitivity. A mediation model indicated punishment sensitivity completely mediated the relation between functional connectivity within a caudate–bilateral MFG network and procrastination. Our results support the theory that those with higher punishment sensitivity have weaker effective emotional self-control supported by the caudate–MFG network, resulting in greater procrastination.

## 1. Introduction

Procrastination is the act of putting off an intended course of action voluntarily despite expecting that the delay will have a large cost in the future [1,2]. Procrastination is a globally prevalent form of self-regulatory failure and is a significant problem for up to 15–20% of adults. Over 95% of procrastinators wish to reduce it [1]. The temporal difference model of procrastination [3] postulates that procrastination results when the temporally discounted positive utility of the task incentive is outweighed by the immediate negative utility. In the present study, we studied individual differences in punishment sensitivity, hypothesizing that participants with higher degrees of sensitivity would show greater procrastination. We combined anatomical and functional connectivity analyses, first using voxel-based morphometry (VBM) to identify a portion of the caudate nucleus associated with punishment sensitivity, then resting-state functional connectivity (rsFC) to identify functionally connected networks between the caudate and cortical regions associated with both punishment sensitivity and procrastination. Finally, we used mediation analysis to test how functional connectivity in this network related to punishment sensitivity and procrastination.

### 1.1. The Temporal Difference Model of Procrastination: Theory and Neural Correlates

Our research hypotheses are based on the temporal difference model of procrastination [3,4], which characterizes procrastination as a decision resulting from the consideration of both positive and negative utility associated with task performance. The positive utility includes the reward and positive emotions associated with task performance and goal achievement, whereas negative utility includes aversive aspects of task performance including negative task-related emotions. Importantly, this theory postulates different temporal discounting rates for the positive and negative utility, such that positive utility is processed on a longer timescale and exhibits greater temporal discounting. This results in relative overweighting of the immediate negative utility and predicts that people will procrastinate on a task when the immediate negative utility outweighs the discounted future utility. In support of this theory, Feng, Zhang, Chen, and colleagues performed a series of experiments using different methods to relate procrastination to underlying brain networks. A task-based fMRI study [5] found dissociable neural systems associated with the effects of positive utility and negative utility on procrastination. Procrastination reduction for a higher-value task was associated with increased caudate activity and increased caudate–hippocampal functional connectivity; a procrastination increase following greater task aversiveness was associated with increased insula activity and increased amygdala-insula functional connectivity. Zhang and colleagues [6] also used task-based fMRI and found that the more that participants were able to associate positive outcomes with tasks, the greater their striatal activity and the greater their hippocampal-striatal connectivity. Yang and colleagues [7] identified neuroanatomical and resting state functional connectivity correlates of individuals’ propensity for different types of episodic future thinking in procrastination. The propensity for positive outcome anticipation was correlated with gray matter volume in the dorsolateral prefrontal cortex and functional connectivity between this area and the inferior frontal gyrus and precuneus, whereas the propensity for negative task engagement anticipation was correlated with the hippocampus volume and functional connectivity with the insula. Chen and colleagues [8] found a positive correlation between procrastination and gray matter volume in the insula, anterior cingulate, orbitofrontal cortex, and parahippocampal gyrus, and a negative correlation in the dorsolateral prefrontal cortex. In combination, these four studies using different methods indicate three neural systems in procrastination: An amygdala-insula-OFC network involved in negative task-related emotions, a hippocampal-striatal network supportive of episodic future thinking and task valuation active primarily but not exclusively during positive task anticipation, and a cognitive control system including DLPFC and ACC that is functionally connected to these other systems for top-down control over emotions and task performance.

### 1.2. The Role of Punishment in Theories of Procrastination

In the temporal difference model of procrastination, anticipated punishment is one factor determining the calculation of negative utility. Individual differences in punishment sensitivity should affect the individual’s tendency to procrastinate by affecting the trade-off between negative utility and positive utility [1,3,4]. A high degree of punishment sensitivity results in increased negative emotions associated with a task, and ultimately results in increased procrastination [9,10,11,12]. Other theories of procrastination make similar predictions about the effect of punishment sensitivity on procrastination. The emotion regulation theory [9] proposes that procrastination results from an individual’s desire to avoid negative emotions such as anxiety and pain associated with the procrastinated tasks. This theory would also predict that punishment sensitivity, which is associated with sensitivity to these negative emotions, would be associated with higher degrees of procrastination [13,14].

### 1.3. The Caudate: Nexus between Reward, Punishment, and Cognitive Control of Behavior

We focused on the caudate nucleus as a proposed nexus between value processing for both reward and punishment, and cognitive control exerted via frontostriatal networks. As summarized above, studies performed by Teng, Zhang, and colleagues together found a relationship between the striatum itself as well as functional connectivity between the striatum and hippocampus in reducing procrastination. Much research shows that the dorsal caudate represents the value of behaviors [15,16,17,18] and plays an important role in the acquisition of stimulus–outcome associations [19,20,21]. The caudate is thought to process reinforcement broadly, and caudate activity has been reported in studies of punishment-based learning as well as reward-based learning [22,23].

Most studies have examined activity in the caudate during tasks involving reward and punishment. A few studies have also examined anatomical correlates of individual differences in punishment and reward sensitivity. Ide et al. [24] studied a sample of 11,542 children and adolescents and found reward sensitivity in the ventral striatum extending superior into the caudate, and punishment sensitivity (as measured by the BIS) in portions of the caudate and putamen.

The caudate interacts with other neural systems to support behavioral choice. Two of these networks have been discussed already as part of the neural systems underlying procrastination. One is the caudate interaction with the hippocampal networks supporting memory retrieval and episodic future thinking [25]; this network is thought to support the integration of value with memory in representing the positive utility of the task. The other is the caudate interaction within corticostriatal loops [26] underlying the self-control network, including the dorsolateral prefrontal cortex, inferior frontal cortex, and the anterior cingulate. These networks exert top-down control on emotional processing regions, and arbitrate when multiple behavioral options are supported by different neural systems (for example, in the arbitration between goal-directed and habitual behavior, Kim et al. [27]). In procrastination, this network is thought to reduce negative utility through the top-down control of emotional processing regions such as the insula and amygdala, as well as directly facilitating task performance to avoid distraction.

### 1.4. Punishment Sensitivity and Individual Differences in Procrastination

We chose to directly examine individual differences in punishment sensitivity using the BIS scale, based on the theory of the behavioral inhibition system (BIS). This system is conceptualized as an attentional system that functions to evaluate potential sources of threat and modulate reactions to aversive stimuli [13]. The concept of BIS (punishment sensitivity) is derived from Gray’s reinforcement sensitivity theory (RST), and is related not only to affective states, behavior, and personality, but also related to predispositions for various forms of psychopathology [13,14,28]. High punishment sensitivity as measured by BIS can lead to withdrawal behavior that may lead to aversive or harmful outcomes [13,29,30]. High punishment sensitivity can lead people to avoid situations in which potentially rewarding stimuli are accompanied by the risk of loss, resulting in behavioral inhibition and negative affect [13,14]. Previous studies suggested that self-reported procrastination was associated with high punishment sensitivity [31]. Previous studies have not examined the neural systems underlying the relationship between punishment sensitivity and procrastination, but Gao and colleagues [32] related procrastination to another individual difference measure, trait conscientiousness. They found that high conscientiousness reduced procrastination through facilitative effects on a component of the self-control network (DLPFC) and suppressive effects on a component of the negative emotional network (insula).

### 1.5. The Current Study

In the present study, two hundred and sixty-eight healthy undergraduate subjects were studied using individual difference measures and neuroimaging. The Pure Procrastination Scale (PPS) and Behavioral Inhibition System Scale (BIS) were applied to measure procrastination and punishment sensitivity, respectively. Whole-brain VBM analysis was performed to define the caudate region associated with punishment sensitivity for subsequent analyses. RSFC was then used to identify cortical areas that were functionally connected to the caudate ROI and within which functional connectivity was related to punishment sensitivity. A mediation model was employed to test how this pattern of functional connectivity may mediate between punishment sensitivity and procrastination. We hypothesized that individuals with higher sensitivity to punishment would be more likely to self-report high levels of procrastination. We further hypothesized that gray matter volume in the caudate nucleus would be related to punishment sensitivity, and functional networks connecting the caudate nucleus with other brain regions would be related to both punishment sensitivity and procrastination.

## 2. Materials and Method

### 2.1. Participants and Procedure

In total, 281 right-handed healthy volunteers from Southwest University (China) were recruited for this study. Three were excluded because they did not complete the MRI scan. Ten were removed because of excessive movement (according to the screening criteria: ≤2.0 mm translation in axis and ≤2.0° rotation in axis) [33], leaving 268 participants (196 females) aged 18 to 25 (mean 20.89 years, SEM 1.4 years) for further analyses. All participants had normal or corrected-to-normal vision and were in good physical health. None of them had any mental illness or family history of mental illness. Participants gave written informed consent prior to the start of the experiment, which had been approved by the Institutional Review Board of Southwest University. All the participants completed the MRI scan before the behavioral measures: The Pure Procrastination Scale (PPS) and Behavioral Inhibition Scale (BIS). They were paid for their participation at the end of the experiment. 

### 2.2. Measures

#### 2.2.1. Procrastination

Procrastination was measured by the Pure Procrastination Scale (PPS) [8,34], which is composed of 12 items from three existing procrastination scales (the General Procrastination Scale (GPS; [35], the Adult Inventory of Procrastination Scale (AIP; [36]) and the Decision Procrastination Scale (DPS; [37]). The three PPS subscales enable this scale to measure three facets of procrastination: Decisional procrastination (DPQ, e.g., “I waste a lot of time on trivial matters before getting to the final decisions”), delay in implementation (GPS, e.g., “In preparation for some deadlines, I often waste time by doing other things”), and timeliness/lateness (AIP, e.g., “I find myself running out of time”). Each item on the PPS was scored by subjects using a 5-point Likert-type scale from 1 (very uncharacteristic) to 5 (very characteristic). The total score across all items was calculated, with higher scores indicating greater procrastination. We obtained a Cronbach’s alpha reliability of α = 0.87. This is similar to an earlier result of Steel [38] that reported internal consistency of the PPS of α = 0.92. Both results indicate that the PPS has an excellent internal consistency.

#### 2.2.2. Punishment Sensitivity

Punishment sensitivity was measured using the Behavioral Inhibition Scale (BIS), a subscale of the BIS/BAS scale [13]. The BIS/BAS scale consists of 20 items and is a widely used measure that assesses sensitivity to cues of threat and reward [39]. The Chinese version was revised by Li et al. [40]. It includes 18 items after the deletion of item 1 (“Even if something bad is about to happen to me, I rarely experience fear or nervousness”) and item 18 (“I have very few fears compared to my friends”) from the original instrument. The BIS subscale consists of 7 items, such as “being criticized or accused will make me feel very sad”. It is scored on a 5-point Likert-type scale from 1 (strongly disagree) to 5 (strongly agree). The higher the BIS total score, the higher the individual’s punishment sensitivity. The scale has high construct validity, discrimination validity, and reliability in Chinese adolescents [41]. Cronbach’s alpha was calculated for the overall BIS/BAS scale (0.70) and the BIS subscale (0.59) [41]. In the present study, we found Cronbach’s alpha coefficient was 0.79 for the BIS.

#### 2.2.3. fMRI Data Acquisition

The anatomical and resting-state fMRI images were acquired on a 3.0-T Siemens Trio MRI scanner (Siemens Magnetom Trio TIM, Erlangen, Germany). T1-weighted anatomical images with a high resolution (voxel size = 1 × 1 × 1.33 mm^3^) were acquired by means of a magnetization-prepared rapid acquisition gradient-echo (MPRAGE) sequence (slices = 128; repetition time (TR) = 2530 ms; echo time (TE) = 3.39 ms; flip angle = 7°; 256 × 256 matrix). In addition, to acquire functional images, we used a T2-weighted echo-planar imaging sequence (TR = 2000 ms, TE = 30 ms, flip angle = 90°, resolution matrix = 64 × 64, field of view (FOV) = 200 × 200 mm^2^, 32 slices, voxel size = 3.1 × 3.1 × 3.6 mm^3^). All participants were instructed to stay relaxed, think of nothing, and keep their eyes open during the resting scan. They were also directed to remain physically still. The whole resting scan lasted 12 min, incorporating 360 brain volumes.

### 2.3. VBM Analysis

#### 2.3.1. Preprocessing 

Structural data preprocessing was performed with the software package Data Processing Assistant for resting-state fMRI (DPARSF) (http://rfmri.org/DPARSF, accessed on 29 December 2020) [33]. Preprocessing included the following steps. First, we converted raw data in DICOM format to NIFTI format. Second, to reorient the T1 image, the structural images were manually adjusted to place the anterior commissure (AC) at the origin of the three-dimensional Montreal Neurological Institute (MNI) space. Third, images were segmented into GM, white matter (WM), and cerebral spinal fluid (CSF) [42]. Fourth, the DARTEL algorithm was then used to create a particular group template and a flow field that stores the deformation information. The GM image of the normalized space was matched to the MNI space using affine spatial normalization in the DARTEL toolbox. Then, the images were modulated using the Jacobian matrix determinant to preserve the GM volume within a voxel. Finally, in order to improve the signal-to-noise ratio (SNR), the images were smoothed using an 8 mm full-width at half-maximum (FWHM) Gaussian kernel.

#### 2.3.2. Second-Level Modeling Analysis

VBM statistical analysis was performed using SPM12 (Wellcome Department of Cognitive Neurology, London, UK, http://www.fil.io-n.ucl.ac.uk/spm, accessed on 31 December 2020). In SPM 12, multiple linear regression was performed with the purpose of locating the brain regions related to punishment sensitivity. BIS score was considered the variable of interest in this model. In line with previous studies, age, sex, and global GM volumes were included as covariates [43,44]. Global GM volumes were obtained by the script “get_totals” in MATLAB (http://www.cs.ucl.ac.uk/staff/g.ridgway/vbm/get_totals.m, accessed on 31 December 2019). An absolute threshold for masking of 0.2 was used. T contrasts were utilized to detect voxels, which were significantly associated with punishment sensitivity. Given our a priori hypothesis that the head of the caudate would be associated with punishment sensitivity, and that the goal of the analysis was merely to identify the most appropriate ROI to use for subsequent analyses, a relatively lenient correction for multiple comparisons was performed for the statistical maps using the Gaussian random field (GRF, voxel *p* < 0.05; cluster *p* < 0.01). Given this lenient threshold, any other areas identified in this analysis should be interpreted with caution. 

### 2.4. rsFC Analysis

#### 2.4.1. Preprocessing

The preprocessing for the resting-state fMRI data was also performed using DPARSF [33]. Preprocessing included the following steps. First, raw data in DICOM format was converted to NIFTI format. The initial 10 volumes were discarded owing to the effect of magnetization disequilibrium and the participant’s adaptation to scanning noise. Three hundred and fifty volumes remained, and they were corrected for temporal shifts between slices and corrected for motion. The 31st slice was the reference since it is in the middle of the scan. Then, the individual T1-weighted images were coregistered with the corresponding functional images, and its coregistered images were segmented into GM, WM, and CSF. Then, the images were normalized to the MNI space in 3 × 3 × 3 mm^3^ voxel sizes and smoothed with a Gaussian kernel of 4 mm FWHM to improve the SNR. The residual signal was temporally band-pass filtered (0.01–0.08 Hz) and linearly detrended to obtain low-frequency fluctuation [45,46].

#### 2.4.2. Functional Connectivity Analysis

This part of the analysis was performed with the REST toolbox (http://restfmri.net/forum/REST_V1.8, accessed on 4 August 2021) [47]. This analysis utilized the caudate ROI identified in the VBM analysis described above in which the GM volume correlated with punishment sensitivity. The ROI was extracted and resampled from 1 × 1 × 1.33 mm^3^ to 3 × 3 ×3 mm^3^ for the subsequent functional connectivity analysis. 

In the first-level analysis, voxel-wise functional connectivity was performed to compute the temporal correlations between the average intensity of the BOLD signal within the defined caudate ROI and that in each other voxel across the whole brain. This step was followed by Fisher z transformation for group-level analysis. In the group-level analysis, to determine the relationship between functional connectivity and punishment sensitivity, the correlation was computed between punishment sensitivity scores and the individual-level z-valued functional connectivity maps. The connectivity networks that survived under GRF correction (voxel *p* < 0.001; cluster *p* < 0.05) were defined as areas of the punishment sensitivity network for further functional connectivity analysis. In order to examine the relation between procrastination and functional connectivity in the resulting punishment network, we calculated the functional connectivity correlation with procrastination. Finally, mediation analyses were performed, using PROCESS for SPSS with 5000 bootstrap samples, to determine the relationship between the identified functional connectivity network, punishment sensitivity, and procrastination.

## 3. Results

### 3.1. Behavioral Results

In order to determine the statistical methods used in the subsequent analysis, normality tests were conducted on the scores of BIS and PPS. First, 10,000 Monte Carlo simulations were carried out for the Kolmogorov–Smirnov test. The results suggested these two variables were normally distributed (punishment sensitivity: Kolmogorov–Smirnov Z = 0.770, *p* = 0.593; procrastination: Kolmogorov–Smirnov Z = 0.790, *p* = 0.058), as shown in Figure 1, making them suitable for subsequent statistical parametric analysis methods.

Next, we explored the effects of sex and age on punishment sensitivity and procrastination. The results showed no significant difference between sexes in punishment sensitivity (men: 35.19 ± 7.69, women: 36.99 ± 7.39; *t*(*df* = 264) = 1.864, *p* = 0.063) or in procrastination (male: 20.35 ± 3.23, female: 21.17 ± 3.18; *t*(*df* = 264) = 1.744, *p* = 0.082). No significant correlations were found between age and punishment sensitivity (*r* = −0.032, *p* = 0.607) or between age and procrastination (*r* = −0.058, *p* = 0.348). Therefore, the effects of these demographic covariates were not considered in subsequent behavioral data analyses.

Pearson correlation analysis showed a significant positive correlation between punishment sensitivity and procrastination (*r* = 0.337, *p* < 0.001) as shown in Figure 1. This validates our hypothesis that individuals with high punishment sensitivity have a higher tendency to procrastinate. 

### 3.2. Neuroanatomical Correlates of Punishment Sensitivity

To identify the caudate region sensitive to punishment in our participant sample, VBM analysis was performed. Using punishment sensitivity as the independent variable, and sex, age, and whole brain gray matter volume as covariates, multiple regression tests were performed. The results (see Table 1) indicated that, as predicted, punishment sensitivity was positively correlated with GM volumes in the right caudate (MNI peak coordinates: 13.5, 7.5, 10.5; voxels = 655; peak *t* = 3.622. This area was defined as an ROI for further study. In addition, our analysis revealed two additional regions. These were the bilateral cerebellum (MNI peak coordinates: 4.5, −51, −3; voxels = 966; peak *t* = 3.72), which was positively correlated with punishment sensitivity, and the left fusiform gyrus (MNI peak coordinates: −40.5, −55.5, −12; voxels = 728; peak *t* = −3.340), which was negatively correlated with punishment sensitivity. Given our a priori hypothesis that the head of the caudate would be associated with punishment sensitivity, and that the goal of the analysis was merely to identify the most appropriate ROI to use for subsequent analyses, a relatively lenient correction for multiple comparisons was performed for the statistical maps using the Gaussian random field (GRF, voxel *p* < 0.05; cluster *p* < 0.01). Given this lenient threshold, the other areas (i.e., Cerebellum and Left Fusiform) identified in this analysis should be interpreted with caution. Because of their unclear relationship with punishment sensitivity, these two areas were not included in subsequent analyses and discussions.

### 3.3. rsFC Results

We next conducted an rsFC analysis to identify areas in which functional connectivity with the caudate correlated with punishment sensitivity. We first performed a whole-brain analysis that measured the functional connectivity of each other voxel in the brain with the caudate ROI and tested whether the connectivity correlated with punishment sensitivity. We found punishment sensitivity was negatively correlated with connectivity between the right caudate and right MFG (middle frontal gyrus; MNI peak coordinates: 39, 54, 3; voxels = 41; peak *t* = −0.258, GRF corrected; see Table 2, Figure 2), and left MFG (MNI peak coordinates: −42, 48, 6; voxels = 45; peak *t* = −0.250, GRF corrected; see Table 2, Figure 2). To further explore the relation between this functional connectivity network and procrastination, we examined correlations with our procrastination measure. Findings indicated that functional connectivity between the right caudate and the right and left MFG was significantly correlated with procrastination (right MFG; *r* = 0.151, *p* = 0.014; left MFG; *r* = 0.172, *p* = 0.005). This result suggested that caudate connectivity with bilateral MFG is associated with both punishment sensitivity and procrastination. 

### 3.4. Mediation Analysis

To further examine the relationship between functional coupling, punishment sensitivity, and procrastination, a mediation model was built. We first collapsed the individual right caudate–right MFG and right caudate–left MFG networks into a single caudate–bilateral MFG connectivity measure, which was defined as the independent variable. Punishment sensitivity was defined as a mediating variable, and procrastination as a dependent variable, as shown in Figure 3. The results indicated that punishment sensitivity played a completely mediating role in the relation between the functional connectivity of the right caudate–bilateral MFG and procrastination (indirect effect = 0.095; SE = 0.029; 95% CI = [−0.154, −0.043]). That is, when punishment sensitivity was added as a mediating variable, the effect of the original caudate–bilateral MFG functional connection on procrastination tendency (*β* = −0.180; SE = 0.060; 95% CI = [−0.299, −0.061]) completely disappeared (*β* = −0.085; SE = 0.061; 95% CI = [−0.204, 0.034]). 

## 4. Discussion

In the present study, we investigated the neural systems underlying the relationship between punishment sensitivity and procrastination by using VBM and rsFC analyses. We first verified that punishment sensitivity and procrastination were correlated within individuals, using established measures of each (BIS for punishment sensitivity, PPS for procrastination). At the neural level, we found that punishment sensitivity was negatively correlated with the intensity of functional coupling between the caudate and bilateral MFG. More importantly, a mediation model found that punishment sensitivity plays a completely mediating role in the relation between functional connectivity of the caudate–bilateral MFG and procrastination. 

This study found individuals with a higher sensitivity to punishment have a higher tendency to procrastinate, consistent with our hypothesis and previous research [31,48]. Within the temporal difference decision model of procrastination that motivated our study [3,4], procrastination is conceptualized as resulting from a trade-off between the negative utility of the current task process and the discounted positive utility of future task outcomes. Thus, higher punishment sensitivity results in higher negative utility estimates, and a higher probability that the person will choose to procrastinate. This result can also be interpreted in terms of the short-term mood repair theory [11], which suggests that one who is in a negative mood would tend to delay the task.

Our primary results were that punishment sensitivity was negatively correlated with a positive functional connection between the caudate nucleus and bilateral MFG, such that as sensitivity increases, connectivity decreases. This is consistent with previous research finding anatomical and intrinsic functional connectivity differences in the fronto-parietal network correlated with individual differences in procrastination [49,50]. The caudate-MFG network has been well-established as an important network underlying self-control [51,52,53], in which people must exert executive control over behavior and emotion in service of goal achievement [54,55,56,57]. In the area of emotional control, the lateral prefrontal system can exert control over subcortical motivational and emotional systems [58,59,60]. It has been found that activity in the lateral prefrontal cortex increases after self-control training [61]. 

Furthermore, results from the mediation model suggested punishment sensitivity plays a completely mediating role in the relation between functional connectivity between the caudate–bilateral MFG and procrastination. In other words, the effect of the caudate–bilateral MFG connection on procrastination can be fully explained by punishment sensitivity. Although procrastination is a complex phenomenon, this mediation model indicates that control over negative emotional processing is key for explaining how frontostriatal connectivity affects procrastination. 

Future research could take into account reward sensitivity in addition to punishment sensitivity. According to the temporal decision theory of procrastination, individuals are affected by both the utility of task execution and the utility of task results, both of which may be related to individual differences in reward sensitivity. In general, performing a task is considered to have a negative utility, whereas the outcome of the task is taken to have positive utility, which constitutes the approach–avoidance trade-off. It is important to note that the effect of reward sensitivity on procrastination may be more complex. Unlike the single role proposed for punishment sensitivity in procrastination (in terms of its effect on negative utility), reward sensitivity may influence procrastination through two pathways. On the one hand, reward sensitivity may increase individuals’ expected reward utility for the future completion of tasks and the outcome of tasks, thus increasing task–approach behavior and reducing procrastination. On the other hand, reward sensitivity may also increase the value of immediate reward stimuli, causing individuals to turn to immediate gratification and forego goal-oriented behaviors, leading to procrastination. 

In the current study, we used a VBM-constrained, seed-based rsFC analysis. This method has been well established [4,62,63,64]. Using this method, Zhang et al. [4] identified neural substrates of trait anxiety responsible for delay discounting. VBM-constrained, seed-based rsFC analysis has two important advantages over previous ways of examining rsFC. First, using VBM helps avoid the biased selection of seeds for rsFC analysis. Second, using VMB allows researchers to extract brain areas related to a particular individual difference without the need for prior information about possible locations. VBM-constrained rsFC analysis was effective in the current study in the identification of punishment sensitivity-specific areas in the caudate in the absence of a strong hypothesis about a specific location. However, one limitation of this approach is that a relatively large sample size is required, especially for extracting seed regions using VBM.

## 5. Conclusions

In summary, we employed multimodal neuroimaging analysis on a large sample to identify neural systems underlying the relationship between punishment sensitivity and procrastination. We found that punishment sensitivity and procrastination share a common neural basis within the network connecting the caudate nucleus and middle frontal gyrus. Most importantly, the effect of the strength of functional connectivity within this network on the propensity to procrastinate was fully mediated by punishment sensitivity.

## Figures and Tables

**Figure 1 brainsci-12-01163-f001:**
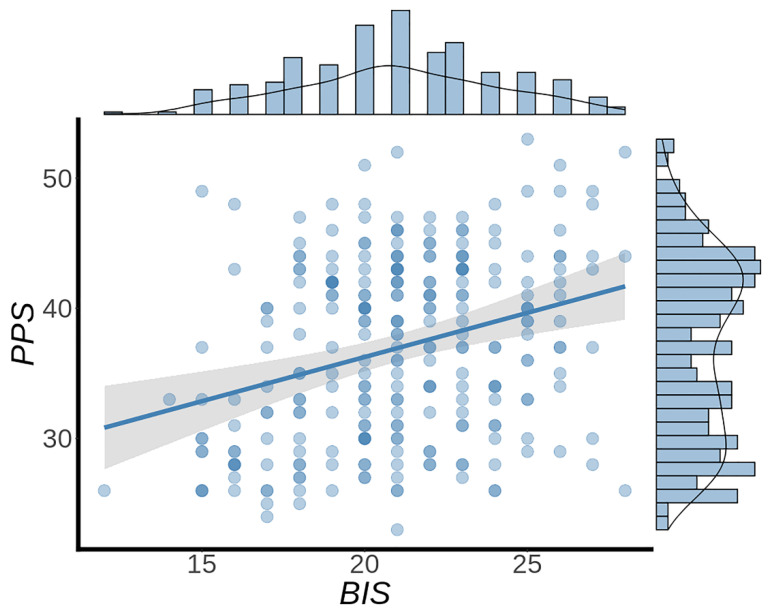
Behavioral results. Punishment sensitivity (BIS, abscissa) was positively correlated with procrastination (PPS, ordinate) (*r* = 0.337, *p* < 0.001). Blue dots indicate values from individual participants. The blue histograms on the top and right side show the frequency distribution of BIS and PPS, respectively. BIS: Behavioral Inhibition Scale. PPS: Pure Procrastination Scale.

**Figure 2 brainsci-12-01163-f002:**
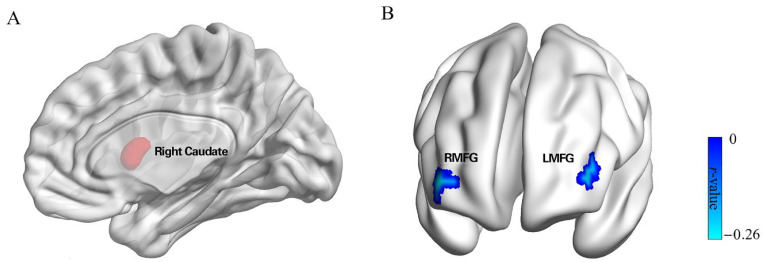
Resting-state functional connectivity networks. (**A**) The right caudate seed region ROI, identified via the VBM results (See Table 1). (**B**) Significant functional connectivity between the right caudate seed region and bilateral MFG (GRF corrected; voxel *p* < 0.001; cluster *p* < 0.05) that was also significantly negatively correlated with punishment sensitivity.

**Figure 3 brainsci-12-01163-f003:**
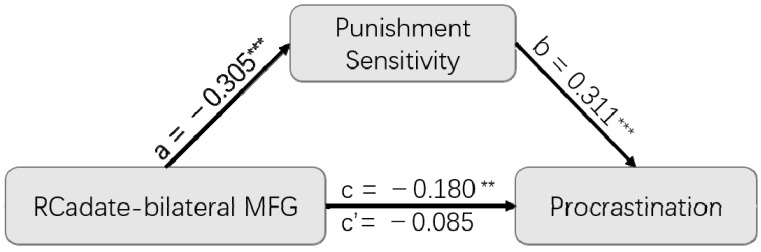
Mediation results. Punishment sensitivity played a completely mediating role in the relation between functional connectivity between the right caudate and bilateral MFG and procrastination. ***: *p* < 0.001; **: *p* < 0.01.

**Table 1 brainsci-12-01163-t001:** VBM analysis results.

Seed	Region	MNI			Voxels	Peak Value
		x	y	z		
Punishment Sensitivity	Bilateral Cerebellum	4.5	−51	−3	966	3.725
	Right Caudate	13.5	7.5	10.5	655	3.622
	Left Fusiform	−40.5	−55.5	−12	728	−3.34

Note: Table shows brain structures identified in the voxel-based morphology analysis in which gray matter volume correlated with punishment sensitivity. The caudate area was predicted a priori and was extracted to use as an ROI in future analyses. The remaining areas should be interpreted with caution given the lenient statistical threshold used (GRF corrected; voxel *p* < 0.05; cluster *p* < 0.01).

**Table 2 brainsci-12-01163-t002:** Functional connectivity networks for punishment sensitivity.

Seed	Region	MNI			Voxels	Peak Value
		x	y	z		
Right Caudate	Right MFG	39	54	3	41	−0.258
	Left MFG	−42	48	6	45	−0.250

Note: Regions listed are those that showed significant correlation in activity with the right caudate (GRF corrected; voxel *p* < 0.001; cluster *p* < 0.05), which also were significantly correlated with punishment sensitivity.

## Data Availability

The data presented in this study are available on request from the corresponding author.

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
