# Peer review of "Frontostriatal Functional Connectivity Underlies the Association between Punishment Sensitivity and Procrastination"

_brainsci, 2022, doi:10.3390/brainsci12091163_

Round 1
Reviewer 1 Report
The authors present a rigorous and high-powered analysis correlating functional connectivity with theoretically-motivated behavioral measures of procrastination and sensitivity to punishment. The methods feel appropriate, for the most part, and the analyses are well aligned with the hypotheses and motivating theoretical model.
I do have comments on the VBM. It seems to me the authors should either define the caudate ROI anatomically or use an established functional localizer. Although prior work showed that parts of the caudate were morphologically correlated with sensitivity to punishment, the introduction does not establish a clear hypothesis about sub-regions of the caudate and it is not clear why using only the morphologically-correlated parts of the caudate in the functional connectivity analysis is necessary or more insightful than an ROI that includes the whole caudate. Also, it is not clear why, if the VBM was only meant to define a caudate ROI, that a whole-brain statistical investigation was performed at a relaxed statistical threshold that can only be interpreted cautiously if at all.
I found myself wondering whether the morphological variability in the caudate correlated with the functional connectivity between that region and frontal cortex.
The mediation, however, nicely shows that the degree of connectivity between frontal and caudate regions is negatively correlated with sensitivity to punishment, and that the connectivity does not predict procrastination after accounting for sensitivity. This is evidence of neural mechanisms in line with the motivating hypotheses.
Reviewer 2 Report
Thank you very much for the opportunity to review this article.
In their work, the authors investigate the neuroanatomical relationship between punishment sensitivity and procrastination.
Overall, it is a very interesting and well-written article.
My only comment:
Line 259 “No significant correlations were found between age and punishment sensitivity (r = .337, p < .001)…”
Is the p value correct?
Author Response
The reported p-value in line 259 was wrong. It should be "No significant correlations were found between age and punishment sensitivity (r = -.032, p = .607)," Please refer to the Result section (P.6, line 251). We thank the reviewer for catching this error.